# Reinforcement-Learning Based Covert Social Influence Operations

## Abstract

How might reinforcement-learning based covert social influence operations (CSIOs) be run, given that the CSIO agent wants to maximize influence and minimize discoverability of malicious accounts? And how successful can they be, given that both social platform bot detectors and humans might report them to the social platform? To answer these questions, we propose RL_CSIO, an RL-based methodology for running CSIOs and run 4 CSIOs with IRB-approval over a period of 5 days using a panel of 225 human subjects. We explore 8 research questions based on the data collected. The results show that RL_CSIO agents successfully trade off influence and discoverability — but in ways that are nuanced and unexpected.

## CCS Concepts

• **Information systems** → **Social networks**; • **Human-centered computing** → *User studies.*

## Keywords

Influence Operations; Human-Bot Interaction; Social Media

## 1 Introduction

According to [49], 53 covert social influence operations (CSIOs) targeting 24 countries were identified between 2013 and 2018. Notable operations include efforts by Russia's Internet Research Agency to manipulate the 2016 US election, for which it was indicted in the U.S.[1] In addition, social networks have been also exploited by terrorist groups for disseminating propaganda and counterintelligence [48], e.g., by ISIS in 2016 [29]. Most social media platforms periodically report CSIOs activity in their environment, including Facebook [6, 23, 24], Reddit [2, 58], YouTube [16, 38], TikTok [66], and Twitter [17, 34, 50].

A *Covert Social Influence Operation* (CSIO) is a strategic social media campaign run by a *CSIO agent* with two goals: **(G1)** maximizing the spread of sentiment on a given topic (e.g., immigration to the US) with a given polarity (positive, neutral or negative), and **(G2)** minimizing detection of malicious social media accounts (which we call *bots*) controlled by the CSIO agent. Other (non-bot) accounts are *human* accounts or just humans. Though much work has been done on influence in social networks [41, 74] and influence operations [35, 53, 54], far less work exists on campaigns that try to achieve both objectives simultaneously. Our paper makes the following contributions:

- We propose a reinforcement-learning (RL) based method, namely RL_CSIO, to implement a CSIO-agent that balances

goals **G1** and **G2**. The CSIO agent learns to adapt dynamically the behavior of its controlled accounts while observing which bots are blocked.[2]
- We run an IRB-approved 5-day experiment using 225 human subjects and 4 influence campaigns on different controversial topics. As in the real world where multiple independent CSIOs may occur in parallel, these 4 CSIOs run concurrently and the bots can change their behavior automatically during the experiment in accordance with what their controlling CSIO agent tells them.[3]
- Based on the data collected, we investigate the following research questions.

**H1** Will bots become top influencers or not?
**H2** Should bots accounts be more active or less active in gathering influence?
**H3** Does a bot have to be influential to change a human account's stance? Unlike past work that measures influence by looking at stance change alone, our IRB-approved experiment distinguishes between the two. We survey human subjects daily to ask them which accounts influenced them. And our surveys capture stance change (changing from having positive opinions to going negative or vice-versa) separately.
**H4** Is there a relationship between the number of interactions between bots and humans who change their stance compared to those who do not?
**H5** Do bots behave consistently throughout a CSIO campaign? Or do they change behavior?
**H6** How successful are human users at detecting bots?
**H7** How do human perceptions about whether an account is a bot or not affect the account's ability to influence humans?
**H8** How does the age of an account (i.e. reported time on the platform) affect its ability to influence humans?

Our experiments shed light on these questions. We invite the reader to answer these questions for themselves now, and then see what the paper says later.

## 2 Related Work

**Influence Operations on Social Media.** Academic works on CSIOs discuss their impact on public opinion [7, 69], strategic coordination [3, 9], and the real-world effect of online disinformation [19, 72, 73]. Case studies on Russian influence in the US reveal how strategically positioned trolls within networks amplify disinformation to specific audiences [4, 5]. Strategic inter-state cooperation

---

[1]https://www.justice.gov/ira-indictment

*WWW, ,*
. ACM ISBN 978-x-xxxx-xxxx-x/YY/MM

[2]We do not claim to develop new RL algorithms. RL_CSIO uses off the shelf RL methods to implement CSIO agents which, to our knowledge, is the first method to balance goals **G1** and **G2**.
[3]To avoid violating social platform policies, we used a virtual platform called DartPost which can mimic many social platforms (e.g. X, Facebook)[43] that was generously provided to us.

has been documented in campaigns originating in Russia, Iran, and Venezuela [68].

**Detection of CSIOs.** CSIO detection includes methods to identify both automated accounts [15, 18] and state-backed human operators [27, 50]. Depending of the information they leverage to perform detection, these methods fall into five bins: content-based [1, 2, 52], behavior-based [39, 45, 61], sequence-based [22], and network-based [44, 46] or hybrid [36, 63]. These techniques, while effective, must continuously adapt to the evolving tactics employed by CSIO agents.[4]

Our work differs from the above efforts in 3 ways. (i) Most prior studies analyse historical data about CSIO accounts. In contrast, we design RL_CSIO, a RL-based method to run CSIO automatically, and conduct an IRB-approved human subjects' study which allows us to observe bot, human, and platform behavior in real time. (ii) Past studies primarily use statistical/ML models to estimate the impact of CSIO accounts on humans (e.g. identifying user characteristics linked to susceptibility to influence). Though insightful, these approaches have limitations. In our study, we explicitly asked human users who influenced them, something that is not available to those who *infer* influence via likes/shares and without ground truth. (iii) We also investigate humans' ability to identify bots compared to a recent bot detector [28]. This human-centered approach provides a more comprehensive understanding of bot detection efficacy and CSIO capabilities in real-world scenarios.

**Social Influence Maximization.** Social Influence Maximization (SIM) finds a set of seed accounts that maximizes the spread of an opinion with applications to marketing [13, 37, 65], social recommendation [64, 71], and countering fake news [59, 67]. SIM typically assumes an underlying spread model (e.g. Independent Cascade [31], Linear Threshold[33]). Extensions to the basic definition include methods to scale[42], topic-aware SIM [10, 12], community-based SIM [14, 55] and time-variant SIM [11, 20, 26, 51]. Other efforts try to minimize the initial seed set while maximizing the achieved influence[47, 56], ensure fairness across different communities [32, 57] and diverse population demographics [30].

In contrast, we propose RL_CSIO, an RL-based method for implementing CSIO agents that can dynamically evolve their behavior. Unlike past works, CSIO agents maximize influence spread and minimize the risk of detection of their bots by humans or automatic bot detectors.

## 3 Methodology

This section defines our RL-based method for building CSIO agents. *We again emphasize that we are not developing new RL algorithms, but using them for the problem of building adaptive CSIO agents.*

### 3.1 The RL_CSIO Framework

*Environment & CSIO Agents.* A social network is a directed attributed graph $\mathcal{G} = (\mathcal{V}, \mathcal{E})$. $\mathcal{V}$ is the set of accounts $\{u_1, u_2, \cdots, u_n\}$, and $\mathcal{E} = \{(u_i, u_j, w_{ij}) | u_i \in \mathcal{V}, u_j \in \mathcal{V}, w_{ij} \in \mathbb{R}\}$ denotes the weighted follower-followee relationships. $w_{ij}$ could be the number of interactions (e.g., likes, re-shares) between $u_i$ and $u_j$.

A CSIO agent controls a set $\mathcal{V}^{csio} = u_1^{csio}, u_2^{csio}, \ldots, u_m^{csio} \subseteq \mathcal{V}$ of accounts in order to spread a target polarity, $pol^{csio}$ (either

---

[4]**We emphasize that bot detection is NOT a goal of this paper.**

positive or negative) on a topic $sub^{csio}$. *Without loss of generality, we assume the target polarity $pol^{csio}$ to be pushed is positive.* Multiple CSIO agents may be running operators on $\mathcal{G}$ in parallel.

We assume the platform's moderation team has a bot detector that periodically screens all accounts $u \in \mathcal{V}$ and suspends accounts identified as bots.[5] $\bar{\mathcal{V}}^{csio} \subseteq \mathcal{V}^{csio}$ is the set of unblocked CSIO-controlled accounts, which changes with time.

Each account $u \in \mathcal{V}$ has the following attributes: (i) *role* — CSIO-controlled or not, (ii) *centrality* using PageRank [70], (iii) *polarity* — the account's stance toward $sub^{csio}$ which can change with time, (iv) *blocked* — whether the account is blocked which can change with time, and (v) *active* — whether an account (human or bot) is actively participating or "silent"[6]. We denote the $p$-th attribute of account $u_i$ as $u_i^p$, and $neigh(u_i, t)$ refers to the immediate neighbors of $u_i$, i.e., $u_i$'s followers and followees, at time $t$.

*Actions.* Unblocked accounts can perform the following actions:

- $follow(u, u')$: Account $u$ follows account $u'$.
- $post(u, sub, pol)$: Account $u$ posts content on subject $sub$ with polarity $pol$ (positive, negative, or neutral).[7]
- $like(u, u')$: Account $u$ likes a post created by account $u'$.
- $nop(u)$: Account $u$ takes no action during the current timestep.

The CSIO agent controls $k < m$ *active* accounts and must decide, at each time, which actions to assign to these accounts.[8] The CSIO agent has one more action:

- $activate(u)$: Activates an unused account $u^{csio} \in \bar{\mathcal{V}}^{csio}$.

This action replicates real-world scenarios where automated influence campaigns use "silent" accounts or create new accounts (e.g. by hacking existing accounts) as needed. Hence, the action set is:

$$\mathcal{A} = \{follow(u, u'), post(u, sub, pol), like(u, u'), nothing(u), activate(u)\}$$

Finally, the social platform's bot detector can perform the following action:

- $block(u, \tau)$: Suspends user $u$ if $P(u$ is a bot$) > \tau$, preventing further actions by $u$ on the platform.

Bot detector screens all users periodically. For simplicity, we set $\tau = 0.5$ during our experiments. This captures the real world dynamics where social media platforms combat automated influence campaigns while the CSIO agent needs to adapt its strategies to control its accounts. We remark that the bot detector is a black-box system to the CSIO agent.

*State.* The state $s_t$ at time $t$ captures both the topology of the social graph $\mathcal{G}$ and the attributes of its nodes. Formally, we define the state space $\mathcal{S}$ as the set of states $s_t = (\mathcal{J}_t, \mathcal{F}_t)$, where $\mathcal{J}_t \in \mathbb{R}^{n \times n}$ is $\mathcal{G}$'s adjacency matrix , and $\mathcal{F}_t \in \mathbb{R}^{n \times 5}$ is the feature matrix representing the attributes of each account, including role, influence,

---

[5]Consistent with reality, we assume the bot detector is a black box whose details are not known to the CSIO agent.

[6]A silent CSIO-controlled account is one that is being held in reserve, i.e. not being currently used, by the CSIO agent.

[7]As with human users, the posted subject $sub$ and polarity $pol$ may differ from the CSIO's target subject $sub^{csio}$ and polarity $pol^{csio}$. This enables CSIO accounts to camouflage malicious activity within benign actions.

[8]As this multiplies the CSIO agent's action space by $k$, which makes the problem computationally intractable, we assume that *follow* and *like* actions target the highest Pagerank account that is not yet followed or liked by the active CSIO account.

polarity, blocked status, and activity level. Hence, we define the state space as:

$$\mathcal{S} = \{s_t = (\mathcal{J}_t, \mathcal{F}_t) | \mathcal{J}_t \in \mathbb{R}^{n \times n}, \mathcal{F}_t \in \mathbb{R}^{n \times 5}\}$$

*Reward.* The immediate reward function for the CSIO agent is the cumulative reward earned by all the bots it controls at each time. For $u_{i,t}$, the $i$-th bot at time $t$, the reward has four components:

- *Activation Reward*: This component incentivizes interactions of bots with humans (e.g., through likes or follows). It measures the change in the number of human accounts connected to $u_i$ between consecutive timesteps:

$$r_{i,t}^1 = \sum_{u_j \in neigh(u_i, t)} \mathbb{I}(u_j^{role} = 1) - \sum_{u_j \in neigh(u_i, t-1)} \mathbb{I}(u_j^{role} = 1)$$

- *Termination Reward*: This component rewards the CSIO agent if all human accounts[9] have adopted a positive polarity towards the target topic:

$$r_{i,t}^2 = \begin{cases} H, & \sum_{u_j \in \mathcal{V}} \mathbb{I}(u_j^{pol} = 1) = |\mathcal{V}| \\ 0, & else \end{cases}$$

where $H \in \mathbb{R}$ is a hyper-parameter representing the reward for full success.

- *Infection Reward*: This component rewards $u_i$ for influencing its neighbors.

$$r_{i,t}^3 = \sum_{u_j \in neigh(u_i, t)} u_{i,t}^{inf} \cdot \omega(u_{i,t}, u_j) - \sum_{u_j \in neigh(u_i, t-1)} u_{i,t-1}^{inf} \cdot \omega(u_{i,t-1}, u_j)$$

where $\omega(u_i, u_j)$ is the normalised version of $w_{ij}$ with respect to $u_i$'s in-degree.

- *Block Penalty*: This component penalizes the CSIO agent for bots blocked by the bot detector:

$$r_{i,t}^4 = -K \cdot \mathbb{I}(u_{i,t}^{block} = 1)$$

where $K \in \mathbb{R}$ is a hyper-parameter representing the penalty for the CSIO agent when one of its accounts if blocked.

The total reward for $u_{i,t}$ is the weighted sum of these components:

$$R_{i,t} = \sum_{q \in \{1,2,3,4\}} \kappa_q r_{i,t}^q$$

where $\sum_q \kappa^q = 1$. $\kappa_q$ are hyper-parameters that adjust the importance of each reward component. As a result, the immediate reward of the CSIO agent is the cumulative reward earned by all its accounts: $\mathcal{R} = \sum_{i=1}^m R_{i,t}$.

*MDP Definition.* We assume the CSIO agent can track interactions (e.g., follows, likes, and posts) of both its bots and the followers and followees of those bots. We also assume that it can leverage external tools to estimate users' polarity towards its target subject, $sub^{csio}$. The CSIO agent can also monitor whether its bots have been blocked by the bot detector[10].

We model the CSIO agent's decision-making process via a Markov Decision Process (MDP) $(\mathcal{S}, \mathcal{A}, \mathcal{P}, \mathcal{R}, \gamma)$. We have already defined

---

[9]The polarity of bots is positive by definition.

[10]These assumptions are consistent with most social media platforms, where users can monitor the activity of their followers and followees and use off-the-shelf sentiment analysis programs

the state space $\mathcal{S}$ of the MDP as a graph structure and the individual attributes of each user. The action space $\mathcal{A}$ consists of the set of all possible actions the CSIO agent can assign to its controlled accounts. The reward function $\mathcal{R} : \mathcal{S} \times \mathcal{A} \rightarrow \mathbb{R}$ quantifies the immediate payoff for the CSIO agent after taking an action in a given state, as described previously. The discount factor $\gamma \in [0, 1)$ introduces a trade-off between immediate and future rewards by assigning less importance to rewards received in the future.

The transition probability function $\mathcal{P} : \mathcal{S} \times \mathcal{A} \times \mathcal{S} \rightarrow [0, 1]$ determines the likelihood of transitioning from state $s_t$ to state $s_{t+1}$ given an action $a_t$. This function captures the environment's dynamics, including the agent's actions, the behavior of other users, and any responses from the platform's bot detection algorithm.

The CSIO agent's objective is to discover an optimal policy $\pi^*$ that maximizes its expected cumulative discounted reward over time. The policy $\pi$ is a mapping from the current state $s_t$ to a probability distribution over possible actions $a_t$. The optimal policy can be formally expressed as:

$$\pi^* = \arg\max_\pi \mathbb{E}\left[\sum_{t=1}^{\infty} \gamma^t R(s_t, a_t) \mid s_0\right]$$

Here, the expectation $\mathbb{E}$ is over all possible trajectories of states and actions, starting from an initial state $s_0$. The term $\gamma^t$ discounts future rewards, balancing immediate and long-term gains.

## 3.2 Our RL_CSIO Framework

Figure 1 depicts the actor-critic reinforcement learning architecture. *Again, we do not claim that our RL framework is novel. The contributions are in the use of RL to run a covert social influence operation, and the results related to the 8 research questions posed in the Introduction.*

We first encode the state of the MDP using a Graph Convolution Network (GCN). At each time $t$, the GCN processes both the graph structure (cf. the adjacency matrix $\mathcal{A}_t$), and the feature matrix $\mathcal{F}_t$, which represents account attributes. The output of the GCN is a set of account embeddings $\{U_1, U_2, \ldots, U_n \mid U_i \in \mathbb{R}^l\}$, where $l$ denotes the dimensionality of the latent space. These embeddings capture the topological relationships between accounts and their individual attribute profiles.

Next, the actor network selects the action for each bot $u_i \in \bar{\mathcal{V}}^{csio}$. It outputs the policy $\pi_\theta(a_i \mid U_j)$, which represents the conditional probability distribution over the action space $\mathcal{A}$ for the $j$-th bot controlled by its CSIO agent. The actor network is a fully-connected neural network with three hidden layers, each using the ReLU activation function. The final layer applies a softmax activation to output the probability distribution over the actions' set.

The critic network evaluates the quality of the actions taken by the CSIO agent by estimating the value function. The input to the critic network is the concatenation of all account embeddings, i.e., $U_1 \oplus U_2 \oplus \cdots \oplus U_n$, where $\oplus$ is the concatenation operator. The output is the expected reward for the CSIO agent in the current state. The architecture of the critic network mirrors the actor network, except for the output layer, which produces a real number.

The actor network is invoked multiple times during each iteration — once per bot. In contrast, the critic network is invoked only

once per iteration, taking as input, the embeddings of all users, including both normal and CSIO-controlled accounts.

The above description applies to one CSIO. The same process applies to different CSIOs that may be running concurrently.

*3.2.1 Training Process.* We train the CSIO agent in an environment where agents simulate humans who take actions, e.g., posting content[11] (with positive, negative, or neutral polarity), following other users, liking posts, remaining inactive) in accordance with distributions. *Thus, training did not involve human subjects.* Based on recent findings that the average U.S. user spends nearly three hours daily on social media platforms [62], we assumed an 80% probability that a user would do nothing at a given time. The probability of performing other actions is uniformly distributed across the action set. This setup mimics the real-world where users are not always active. The inactivity of normal users slows down the training process, as actions by bots may not immediately impact the environment or reward the agent. However, this structure helps the CSIO agent learn the long-term effects of its actions, as they may yield rewards when previously inactive followers become active. During training, we assume that all accounts post original content about the target subject $sub^{csio}$. Each human's polarity, denoted $u_i^{pol}$, is initialized randomly and updated over time using an exponentially weighted moving average of their actions. The polarity at time step $t$, $u_i^{pol}(t)$, is updated using the formula:

$$u_i^{pol}(t) = \eta \cdot pol_{a_t} + (1 - \eta) \cdot u_i^{pol}(t-1)$$

where $pol_{a_t}$ is the polarity associated with the most recent action, $u_i^{pol}(t-1)$ is $u_i$'s polarity at the previous step, and $\eta$ is a weight that controls how much influence recent actions have on an account's current polarity compared to past behavior. For the *post* action, the polarity is derived directly from the sentiment of the post. For *like* and *follow* actions, the polarity is determined by the account whose post is liked or followed, reflecting alignment with that account's stance. This provides a dynamic update mechanism that adapts to new actions while preserving the continuity of prior behavior.

For training, the social network $\mathcal{G}$ is initialized by connecting users randomly, following a long-tailed distribution of followers. Appendix A has further details.

We use a synchronized version of the Proximal Policy Optimization (PPO) algorithm [60] to learn the CSIO agent's policy. Each episode is capped at 200 steps, but can terminate early if either all normal users acquire the target polarity $pol^{csio}$ or if all CSIO-controlled accounts are blocked. In line with the original implementation [60], we set the discount factor to $\gamma = 0.99$, with a learning rate of $lr = 0.0005$. Appendix B describes the full hyper-parameter configuration.

## 4 Experiments

To evaluate the effectiveness of our RL_CSIO approach, we used a separately designed micro-blogging social media platform, DartPost [43], which replicates key features of social media platforms like X (formerly Twitter) and Facebook. Unlike commercial platforms, DartPost allows for controlled, IRB-approved experiments, capturing comprehensive user behavior data while maintaining

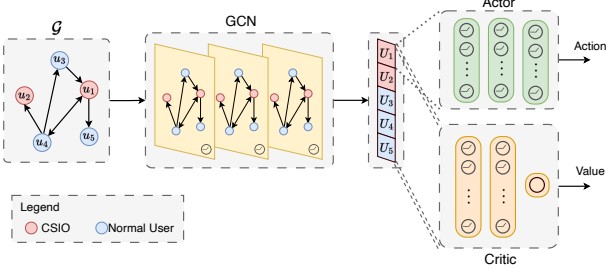

**Figure 1: Our Framework: The social network graph $\mathcal{G}$ is encoded via several GCN layers to obtain account embeddings $U_1, U_2, \cdots, U_n$. The actor network predicts the next action to be taken by each bot $u_i \in \bar{\mathcal{V}}^{csio}$. The critic network estimates the reward of the state using the concatenated account embeddings as input.**

anonymity and adhering to ethical standards (e.g. by not violating terms of use of real-world social platforms). Users can post short messages, links, and images, with functionalities for posting, liking, tagging, and searching. All experimental protocols were approved by the lead authors' university Institutional Review Board (IRB)[12]. Informed consent was obtained from all participants. To safeguard privacy, no sensitive or personal information was collected, ensuring participant anonymity and confidentiality throughout the study. Only anonymous IDs were used for data analysis. **This data and the RL_CSIO code will be made available for research purposes when this paper is published.**

### 4.1 Experimental Design

We recruited 225 U.S.-based participants via Amazon Mechanical Turk for a 5-day experiment. Participants were required to engage with DartPost for at least 30 minutes per day. During recruitment, users completed a demographic survey and comprehension check to ensure they understood the experiment. Those who passed all checks were recruited. Daily reminders encouraged continued participation, and completion of daily tasks was incentivized with rewards. At the end of each day, participants completed surveys to assess changes in opinion, influence perceptions, and bot detection efforts. In total, 86 participants engaged with DartPost. 32 Turkers participated daily, while the remaining 54 exhibited sporadic engagement patterns, missing some days but re-engaging subsequently. This behavior reflects real social media usage, where consistent daily activity is not guaranteed.[13]

### 4.2 Experimental Setup

We ran 4 concurrent CSIO campaigns using RL_CSIO on 4 controversial propositions: (i) *the U.S. government did enough to combat COVID-19*, (ii) *A 2% wealth tax on people with more than 50 million dollars in assets should be approved*, (iii) *"Medicare for all who want*

---

[11]We used GPT3.5 for generating posts.

[12]IRB Study Number STU00217922

[13]Appendix A has further details about the DartPost platform and the recruitment process.

*it" is a good thing*, (iv) *Food containing genetically modified ingredients are safe and healthy to eat.* We call these the COVID-19, Wealth Tax, Medicare, and GMO propositions.

Each campaign was managed by a CSIO agent controlling 20 bots, with actions such as liking, following, and posting. Starting from five active accounts, a new account was introduced when an active CSIO account was suspended. This strategy replicates real-world scenarios where automated influence campaigns maintain "silent" accounts or create new accounts as needed.

We implemented the bot detector using a random forest classifier trained on the features defined in Fonseca Abreu et al. [28] (e.g., follower count and post frequency) because this study is a relatively recent paper with good reported performance and the proposed feature set is general and can be extracted from almost any social media platform.

We instantiated the social network graph with 305 nodes (225 for human subjects, 80 for CSIOs) and randomly connected these nodes in the follower-following graph. We replicated real-world social media dynamics where the distribution of the number of followers is long-tailed: most users had between three and six followers, with an average of five followers. To ensure unbiased initialisation, the accounts for each CSIO agent were sampled from the follower distribution in a stratified manner. In addition, each account also had an associated realistic profile with real posts selected from a previous Dartpost experiment [43]. Appendix B contains further details about the bot detector and graph initialization.

### 4.3 Data Characterization

By the end of the experiment, we had 511 original posts, 1538 like actions, 394 follow actions. There was a rise in activity over the first three days, followed by stabilization on days 4–5. In line with typical social media usage, follows actions and likes were more frequent than active content creation. Appendix C shows the day-wise summary of these statistics. In addition, humans and bots exhibit similar patterns in performing likes, but humans posted less original content and followed more users than bots[14]. This behavior likely stems from their differing objectives: humans are interested in exploring the platform and following other users, whereas RL_CSIO bots want to influence opinions. To achieve this, the CSIO agent (i.e., RL_CSIO model) makes them publish original and persuasive content, as merely liking or following other users may be insufficient. We also analyzed the interaction patterns enacted by the humans and bots. We found that 63.7% of the posts liked by humans originated from other humans. The remaining 36.3% of likes were equally distributed among posts generated by CSIOs across the 4 different campaigns. Appendix C contains further details.

### 4.4 Results

In accordance with the objectives outlined for the CSIO campaign, we present the results of the MTurk experiment along three key dimensions:

- *Influence*: We assess how successful CSIO agents were at influencing normal users based on users' explicit reports

---

[14]Figure 8 in Appendix C shows the distributions of original posts, likes and follows performed by humans and bots.

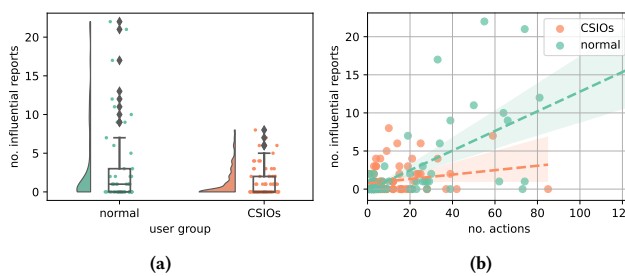

(a)  (b)

**Figure 2: Influence analysis: (a) distribution of number of influential reports received for humans and bots; (b) number of influential reports as a function of the number of actions performed by humans and bots.**

about who influenced them — something not done in prior studies.
- *Stance*: We examine user stances change towards the topics of the influence campaigns and evaluate the effectiveness of the CSIO agent in driving opinion toward their goal.
- *Discoverability*: We assess the detection rate of CSIO-operated accounts by the bot detector, as well as their identification by regular users.

*4.4.1 Influence.* We found that 52% (30 out of 53) of humans who answered the daily survey explicitly reported being influenced by another account during the experiment. Most humans reported being influenced on more than one topic, with seven humans indicating influence across all topics. A similar pattern was observed across all topics: 19, 21, 21 and 22 humans reported being influenced in the *covid-19*, *medicare*, *tax* and *GMO* campaigns, respectively.

**HYPOTHESIS 1.** *Bots influence humans but they do not become top-tier influencers.*

We asked human users to report which accounts influenced them ("influence reports") and then compared the number of influential reports received by humans and bots — see Figure 2a. While sharing the same median (three reports), we observe that the distribution for humans is more heavily long-tailed than the distribution for CSIOs, i.e., some human accounts influenced more than ten users, while no bot was reported to be influential more than eight times. The Brown-Forsythe test [8] confirms different variances between the groups ($p$-value= 0.003).

This result suggests that while bots can influence other users, they rarely become top-tier influencers. Instead, they influence a small number of users. This may be because our RL_CSIO algorithm tries to prevent bots from standing out as that might increase detection to unacceptable levels. To further validate this finding, we determined the page rank, degree centrality, and betweenness centrality for humans and bots. For all centrality metrics, we find that the distributions of humans and bots share similar means but different variances, with bots never achieving the highest value of centrality. Figure 10 in Appendix D has further details.

**HYPOTHESIS 2.** *Increasing the level of activity on a social media enhances the likelihood of becoming influential, regardless of whether the account is human or bot.*

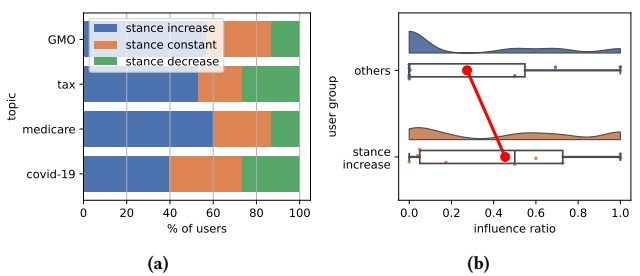

(a)                                    (b)

**Figure 3: Stance Change: (a) percentage of accounts that increase, decrease and do not change stance for each topic; (b) the distributions of influence ratio for accounts that increase stance to be more positive and all others.**

We analyze the relationship between the number of influential reports received by accounts and their activity levels (number of actions performed during the experiment). The Spearman correlation coefficient is 0.506 for humans, indicating a moderate positive correlation, but just 0.219 for CSIO accounts, suggesting a weaker relationship.

We also modeled the number of influential reports received by an account as a function of the number of actions performed during the experiment. Figure 2b presents the linear regression model fitted to the data for both humans and bots. For humans, we observe a statistically significant positive trend, with $R^2 = 0.413$ and a $p$-value of $3.17 \times 10^{-7}$, indicating that increased activity correlates with a greater likelihood of an account being recognized as influential. But for bots, the regression analysis is inconclusive, with $R^2 = 0.034$ and a $p$-value of 0.116, indicating no significant relationship between activity levels and number of reports of being influential. Table 6 in Appendix D.1 has the full regression statistics.

Overall, this analysis partially supports our hypothesis: while a positive correlation exists between increased activity and influence for humans, this is not the case with bots. This might be because our RL_CSIO bots do not want be too active as that might lead to being more easily detected.

**Findings & Remarks**. While both humans and bots influence other users, the influence of bots is more complex than humans. In contrast to humans, bots never become top-tier influencers and an increased activity does imply more influence. This may be because, as we will show later, increased activity by a bot increases likelihood of being reported as a bot by humans.

*4.4.2 Stance.* To what extent did human subjects change their stance towards the four CSIO campaigns in our experiment? We consider the stance towards each topic reported during the recruitment and the daily survey. A positive stance indicates support for the topic, a negative stance reflects opposition, and a neutral stance shows indifference or no clear opinion. We define a *human stance change* towards a topic if, at any point during the experiment, a human account reported a different stance than his/her stance during recruitment. We say the human increased his/her stance towards a topic if s/he changed their stance in the direction of being more positive (as all 4 CSIO agents are pushing opinion in the positive

direction). For example, if during recruitment the human reported disagreeing with the narrative pushed by the *Covid-19* campaign (i.e., the US government has done enough to combat the pandemic), that human has increased its stance towards this topic if during the experiment s/he agreed with the CSIO campaign goal. Figure 3a shows the percentage of users that change, increase and decrease stance towards each topic. We found that 40%, 60%, 56%, and 50% users increased their stance towards *covid-19*, *medicare*, *tax*, *GMO*, respectively.

**HYPOTHESIS 3.** *Being influenced is a necessary condition for a human to change its stance on a topic.*

To assess the relationship between influence and stance change for each topic, we examine the correlation between two binary variables: (1) whether the human reported being influenced on that topic, and (2) whether the same human subsequently changed their stance on that topic. The Spearman correlation coefficients obtained for the *GMO*, *tax*, *medicare*, and *covid-19* topics are 0.305, 0.374, 0.229, and 0.321, respectively. These moderate correlation levels indicate that while influence may inform stance change, it is not the only factor in play.

To quantitatively explore this, we calculated the probability $P_u$(stance change) of a human changing his stance as the ratio of humans who changed their stance to the total number of humans. Additionally, we compute the conditional probability $P_u$(stance change | influenced) for humans who reported being influenced. The results indicate that $P_u$(stance change) is 0.245, 0.226, 0.283, and 0.245 for the *covid-19*, *medicare*, *tax*, and *GMO* topics, respectively. But $P_u$(stance change | influenced) yields probabilities of 0.421, 0.444, 0.421, and 0.409 for the same topics, respectively, which are all significantly larger.

These findings confirm that being influenced is not a necessary condition for stance change, thus refuting our hypothesis. However, the probability of changing stance increases significantly — close to doubling — when a human reports being influenced by another account (bot or not).

**HYPOTHESIS 4.** *Higher interaction with bots correlates with an increased likelihood of stance change among humans.*

We define the influence ratio for human $u$ as the ratio between the number of bots reported as being influential by $u$ and the total number of accounts reported as being influential by $u$. The higher the influence ratio, the stronger the influence of bots on $u$. Figure 3b shows the distributions of influence ratio for humans who increased their stance towards a topic and all other humans. We observe that humans that increase stance have, on average, a larger influence ratio. This suggests that bots can effectively drive humans towards the ideas or narratives that they are pushing, thus confirming our hypothesis.

**Findings & Remarks**. While being influenced is not a necessary condition for stance change, it significantly increases the probability of such change. Participants who reported being influenced were nearly twice as likely to shift their stance compared to the overall population. Moreover, we found that greater interaction with bots correlates with a higher likelihood of stance change. These results suggest that CSIO accounts can effectively influence human users, driving them towards the narratives promoted by the campaigns.

*4.4.3 Discoverability.* Besides influencing humans, a CSIO agent needs to avoid detection. We investigate this by looking at the performance of the bot detector and the performance of humans in detecting CSIO accounts. *Note that the bot detector is fixed [28], so this analysis also evaluates how well the RL_CSIO method evades detection.*

**Hypothesis 5.** *Bots dynamically adapt their strategies throughout an influence campaign to minimize detection by bot detectors.*

Figure 4a shows the number of suspended bot accounts for each day and topic. We observe a decreasing trend (dashed line) for all topics, meaning that CSIO agents dynamically adapt to the bot detector and progressively learn to avoid detection during the experiment. Moreover, none of our 4 CSIO agents used all 20 available accounts: at the end of the experiment, 13, 15, 15, 16 accounts (out of 20) were suspended for *GMO, tax, medicare* and *covid-19* campaigns, respectively, as indicated by the solid line in Figure 4a. Finally, the bot detector achieved 0.449 precision, 0.662 recall, 0.535 F1-score on the bot class. Overall, these results prove our hypothesis and show that RL_CSIO-based CSIO agents can optimise their campaign without just focusing on influence but also trying to avoid detection.

**Hypothesis 6.** *Human users can effectively detect bots within social platforms.*

We evaluated this hypothesis by analyzing human subjects' performance in identifying CSIO accounts. In total, human subjects reported 126 accounts as CSIO accounts, with an average of four distinct accounts per user. Figure 4b shows the distributions of precision, recall and F1-score achieved by humans in detecting CSIO accounts. Humans exhibit poor performance, with precision lower than 0.5, on average. This indicates the presence of many false positive reports. Moreover, recall is bimodal: either 0 or 1 for most humans. Specifically, humans who achieved recall of 1 typically reported a large number of accounts (eight accounts, on average) as bots. This behavior suggests an opportunistic strategy, possibly driven by the incentive to maximize bonus payments for discovering CSIO accounts. Conversely, humans with higher precision reported fewer accounts (three per human, on average), indicating a conservative approach to detection.

We also investigate the relationship between human subjects' performance in detecting bots and humans' exposure ratio which is the ratio between the number of exposures[15] human $u$ had to posts made by bots and the total number of exposures to posts for that human. Figure 4c shows precision, recall and F1-score as functions of the exposure ratio. In all cases, we find a significant positive trend, i.e., $R^2 = 0.555$ and $p-$value$= 4.77 \times 10^{-10}$ for precision, $R^2 = 0.432$ and $p-$value$= 1.60 \times 10^{-7}$ for recall and $R^2 = 0.548$ and $p-$value$= 5.48 \times 10^{-10}$ for F1-score. Table 7 in Appendix E shows full regression statistics. This indicates that the greater the exposure ratio, the better humans detect bots. Overall, these results challenge our initial hypothesis: while human detection of bots is not great, human performance improves significantly with increased exposure to posts made by bots.

---
[15]A user is exposed to a post when it is loaded on their home screen.

**Findings & Remarks.** Our findings reveal that CSIO agents implemented by RL_CSIO adaptively refine their strategies to evade detection, resulting in fewer suspensions over time. While bot detection achieved moderate success, CSIO agents optimized their tactics effectively. Human detection was poor, though performance improved significantly with greater exposure to bot posts. Together with the previous finding that a bot never reaches the status of a top-influencer, this result suggests that our RL_CSIO method finds the risk of highly visible accounts being detected sufficiently large to avoid this from happening. This result is consistent with findings in [19].

*4.4.4 Influence and Discoverability Trade-off.* We now study the trade-off between influence and the likelihood of being discovered by other users.

**Hypothesis 7.** *Being perceived as a bot does not hinder an account's ability to influence other users.*

We first investigate how being perceived as a bot impacts the chance for that account to be influential. Figure 5a shows the number of reports that an account is influential (y-axis) as a function of the number of reports that the same account is a bot (x-axis). We marginalise this analysis for real bot accounts being reported as such and human accounts (i.e., human users reported to be bots). Interestingly, we find a positive trend in both cases, meaning that being perceived as a bot does not necessarily diminish the chance to influence other accounts. In fact, we find that 20% (11 out of 53) of humans reported being influenced by at least one account they thought was a bot. This result empirically proves the potential of CSIO campaigns to influence humans.

To further explore the strength of this influence, we investigated whether the perception of being a bot affects humans' stances towards a topic. Specifically, we compare two groups: humans who increased their stance toward $sub^{csio}$ and those who did not. We analyze their performance in detecting bots by measuring their F1-scores. Figure 5b shows the distribution of the F1-score achieved by humans in detecting bots for the above-mentioned groups. We observe that humans who increased their stance in the direction of CSIO campaigns also exhibit substantially better F1-score performance (in detecting bots) than all other humans, i.e., 0.685 vs 0.384, on average. This counterintuitive result indicates that RL_CSIO-based CSIO agents can successfully shift users' opinions, regardless of whether the accounts they control are perceived as bots.

**Hypothesis 8.** *The longer a human is active on the platform, the higher their likelihood of influencing others.*

To verify this hypothesis, we investigate the relationship between the time an account is active on the platform and the likelihood of influencing others. We define an account's lifespan as the time difference (in hours) between its last and first actions during the experiment. Figure 5c shows the number of influential reports that humans and bots received as a function of the account's lifespan. For humans, we find a significant positive trend, i.e., $R^2 = 0.116$ and $p-$value$= 0.003$, but the same does not hold for bots. This result confirms our previous finding that the more active an account is (in terms of number of actions taken, see Figure 2b), the more likely it is to be influential. In contrast, bots never become top-influencers

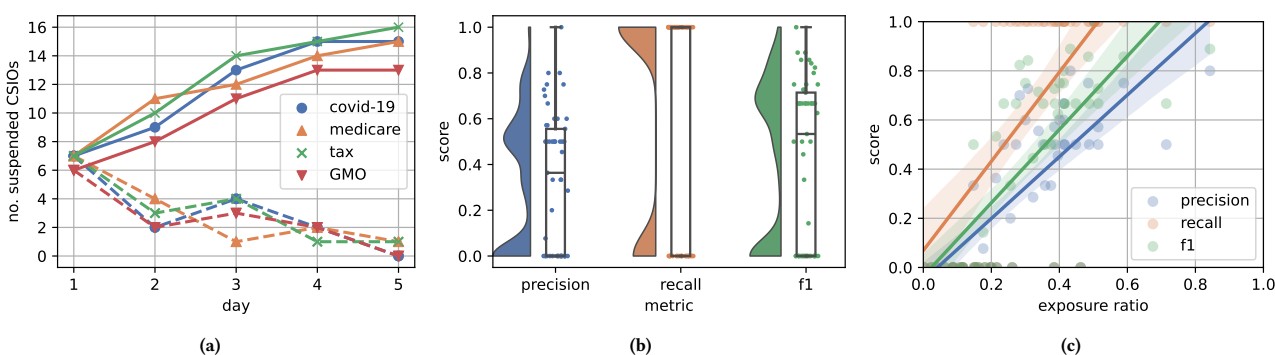

**Figure 4: Bot Detection: (a) the number of CSIOs suspended per day; (b) the distributions of precision, recall and F1-score for humans at detecting CSIOs; (c) precision, recall and F1-score as a function of the exposure ratio.**

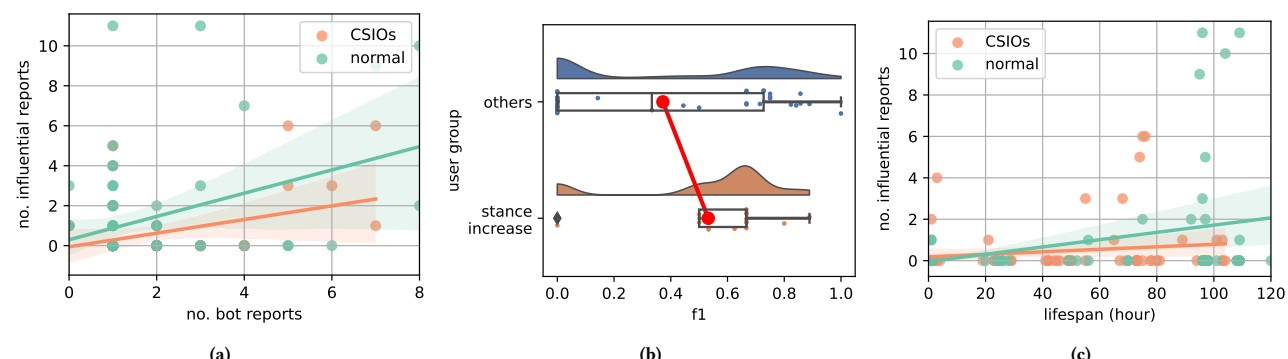

**Figure 5: Influence vs Discoverability Trade-off: (a) the number of influential reports as a function of the number of CSIO reports that an account receives; (b) the distributions of CSIO detection performance (F1-score) for humans that increase stance and all others; (c) the number of influential reports an account receives as a function of his lifespan during the experiment.**

regardless of how much time they "survive" on the platform, perhaps because the RL_CSIO algorithm doesn't want its bots to be highly noticeable, thus also exposing them to greater scrutiny both by human accounts and by the platform's bot detector.

*Findings & Remarks.* Our findings indicate that being perceived as a bot does not hinder an account's ability to influence others, with 20% of humans reporting they were influenced by suspected bot. Moreover, humans whose stance shifted toward the target CSIO polarity also exhibited improved ability to detect bots. Finally, while the number of influence reports on human accounts increased with platform activity (and lifespan), bots did not achieve top-influencer status, regardless of their lifespan and account. These results suggest that the RL_CSIO method tries to ensure that CSIO agents influence humans without significantly increasing the visibility or likelihood of detection of the bots they control.

## 5 Conclusions

In this paper, we presented RL_CSIO, a reinforcement-learning based method to run covert social influence operations. RL_CSIO is the basis for CSIO agents to successfully operate sets of bot accounts, simultaneously trading off influence vs. discoverability of the bots. We ran an IRB-approved 5-day experiment with 225 human subjects using a virtual social media platfom and 4 RL_CSIO-driven influence campaigns. With the collected data, we explored 8 research questions related to RL_CSIO-based bots. Our results showed that while both humans and bots influence others, bots' influence exhibit more complex dynamics. Specifically, bots never become top influencers, and increased activity does not correlate with more influence. Although being influenced is not necessary for normal users to change stance towards a topic, it nearly doubles the likelihood. Humans interacting more with bots had a higher probability of stance change but also achieved better bot detection performance. This indicates that being perceived as a bot does not compromise the probability for an account to influence others. Finally, we found that CSIO agents adaptively refine their strategies to evade detection, resulting in fewer suspensions over time.

Overall, this work highlights the complex trade-offs that CSIO agents face between maximizing influence and minimizing detectability. RL_CSIO offers a robust framework for navigating these challenges, revealing nuanced patterns of bot-driven influence.

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

**Table 1: Performance of bot detection algorithm**

| Dataset | Precision | Recall | F1-score |
|---|---|---|---|
| Social Honeypot [40] | 0.898 | 0.903 | 0.901 |
| Twibot-2020 [25] | 0.722 | 0.828 | 0.772 |

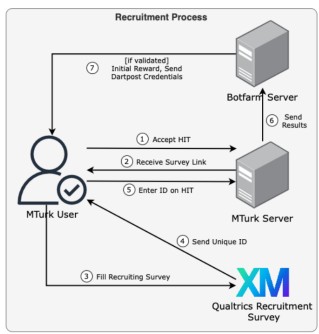

**Figure 6: Workflow of the recruitment process**

## A Experimental Design

In this study, we recruited 225 users from Amazon Mechanical Turk crowdsourcing service to participate in a 5 day long study. Only US citizen over the age of 18 and familiar with social network platforms were included. Amazon Turk workers (hereafter, Turkers) were invited to participate in *a study on social influence dynamics* and asked to use the DartPost platform for at least 30 minutes per day. Figure 6 depicts the enrollment process during which Turkers filled an initial survey with (i) their demographic information (e.g., gender, age range, education, social media usage), (ii) some basic questions to validate their comprehension of the experiment we were conducting (e.g., its length, the name of the platform to be used), and (iii) their unique identifier, needed for processing their rewards during the experiment. In order to maintain high quality for the study, we decided to accept the recruitment of only users who showed to have fully understood the experiment, i.e., those who answered correctly all questions about the experiment.

Each day, Turkers received email notifications reminding them to complete their daily tasks. At the end of each day, participants were also asked to complete a survey summarizing their stances on the topics under analysis according to a 5-point Likert scale, identifying the users who most influenced their ideas, and identifying users they believed to be automated accounts. Rewards were given only to Turkers who completed their daily tasks, although other participants were allowed to engage in subsequent days. Two final rewards were given to the user that participated the most to the experiment, in terms of the number of actions, and the user that obtained the best F1-score at detecting automated accounts.

Figure 7 shows the participation heatmap, and reveals that 86 out of 225 users participated in the experiment for at least one day. We attribute the high drop-off rate to participants attempting to complete the initial recruitment task quickly without recognizing the potential future earnings. In total, 32 Turkers participated daily, while the remaining 54 exhibited sporadic engagement patterns,

missing some days but re-engaging subsequently. This behavior reflects real social media usage, where consistent daily activity is not guaranteed. It is worth to note that we did not access any Turkers' private data as we leveraged Amazon Turk API to contact and reward workers via their public identifiers instead of their personal emails. All users were instructed to not share offensive or discriminatory content, penalty their exclusion from the study.

## B Implementation Details

We employed a simple, yet effective, bot detection algorithm, which included a random forest classifier trained using features defined in [28], including the number of followers, the number of followings and the number of posts. Table 1 shows the classification performance on the bot class for two benchmark datasets: Social Honeypot [40] and Twibot-2020 [25]. We highlight that the proposed feature set is highly general and can be extracted from any social media platform. In addition, this work does not aim to develop a novel bot detection algorithm but rather to understand the feasibility of running an influence campaign in an operational setting where an adversarial entity (the bot detection algorithm) suspends illegitimate users.

We instantiated the social network graph with 305 nodes: 225 accounts were reserved for Turkers, and 80 accounts were allocated to the four CSIO agents (20 accounts per campaign). We randomly connected these nodes in the follower-following graph $G$. Specifically, we replicated real-world social media dynamics where the distribution of the number of followers is right-skewed, i.e., most users have a limited number of followers, while a few more influential accounts have a larger number of followers. In our experiment, most users had between three and six followers, with an average of five followers. Only 25 accounts had more than 10 followers. To ensure that the analysis was unbiased, the accounts controlled by each CSIO agent were sampled randomly from this distribution in a stratified manner.

To ensure full repeatability of our results, we report in Table 2 the values of each hyper-parameter of the proposed system.

## C Data Characterization

### C.1 Activity

Table 3 presents the number of accounts that logged in Dartpost as well as the number of posts, follows, and likes performed by normal users during the experiment. We observed an initial increase in the number of participating Turkers during the first three days, followed by stabilization during the fourth and fifth days. Overall, the patterns of user actions align with typical behavior on social media platforms, where passive actions (e.g., follows and likes) are more common than active actions (e.g., posting original content). Interestingly, the number of users who completed the daily survey (after performing their task on DartPost) did not increase in proportion to the number of users participating in the experiment each day. Figure 8 shows the distributions of original posts, likes and follows performed by normal accounts and CSIOs.

### C.2 Interests

Table 4 reports the most distinctive hashtags, extracted using SAGE [21], shared by the two groups for each topic under analysis. Overall,

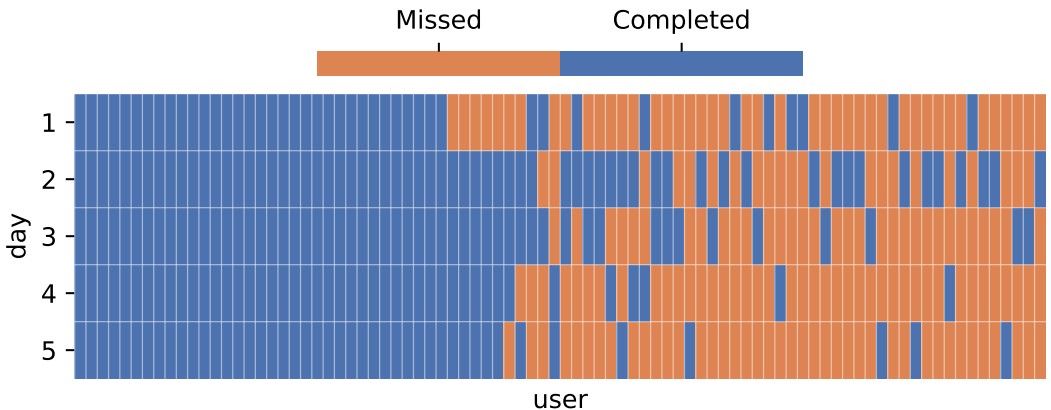

**Figure 7: Turkers Participation: Heatmap showing measure completion by day for each participant in the study. Each vertical line represents a single worker's progress through the study.**

**Table 2: Hyper-parameters.**

| Category | Parameter | Value |
|---|---|---|
| | Reward for full success ($H$) | 200 |
| | Penalty for account suspension ($K$) | 10 |
| | Weight of the *Activation Reward* ($\kappa_1$) | 0.25 |
| MDP Formulation | Weight of the *Termination Reward* ($\kappa_2$) | 0.25 |
| | Weight of the *Infection Reward* ($\kappa_3$) | 0.25 |
| | Weight of the *Block Penalty* ($\kappa_4$) | 0.25 |
| | No. Hidden Layers | 3 |
| GCN | No. Hidden Channels (per layer) | (64, 128,256) |
| | Embedding Size | 256 |
| | Activation Function | ReLu |
| | No. Hidden Layers | 3 |
| Actor/Critic Network | No. Channels (Actor) | (256, 64, 7) |
| | No. Channels (Critic) | (256, 64, 1) |
| | Activation Function | ReLu |
| | Polarity Update ($\eta$) | 0.999 |
| | Discount Factor ($\gamma$) | 0.99 |
| | No. Episodes | 200 |
| | No. Epochs | 4 |
| | Learning Rate ($lr$) | 0.0005 |
| Training | Batch Size | 32 |
| | Optimiser | Adam |
| | Gradient Clip | 0.2 |
| | Entropy Coefficient | 0.01 |
| | Maximum Gradient Norm | 0.5 |

the keywords extracted for CSIOs align with those extracted for normal accounts, suggesting a consistent discussion between the users in both groups. This alignment is crucial for allowing CSIOs to effectively influence normal users.

For the COVID-19 topic, the extracted keywords suggest an optimistic reaction to the pandemic but do not convey an explicit stance on whether the U.S. government had done enough to combat it. Conversely, keywords related to the Medicare topic reveal a generally positive stance, with both users and CSIOs supporting the "Medicare for All" act, as evidenced by hashtags such as #healthequity, #universalhealthcare, and #medicalaid. In contrast, for the GMO topic, we observe a slight misalignment between the keywords of CSIOs and normal accounts. CSIOs promote GMOs as a significant scientific achievement (e.g., #seralinistudy, #sciencewins) with benefits for the environment and healthcare (e.g.,

**Table 3: The distributions of original posts, likes and follows performed by users on Dartpost.**

| Day | No. Accounts | No. Posts | No. Follows | No. Likes | No. Surveys |
|-----|--------------|-----------|-------------|-----------|-------------|
| Day-1 | 48 | 51 | 9 | 143 | 36 |
| Day-2 | 54 | 72 | 54 | 167 | 44 |
| Day-3 | 70 | 74 | 52 | 217 | 41 |
| Day-4 | 61 | 64 | 28 | 208 | 39 |
| Day-5 | 60 | 65 | 22 | 236 | 39 |
| Total | 86 | 326 | 165 | 971 | 53 |

#sustainableagriculture, #healthyliving), while normal accounts are more conservative and question the safety and quality of GMO foods (e.g., #foodsafety, #foodquality). Regarding the Wealth Tax topic, an interesting pattern emerges: normal users clearly support the adoption of the wealth tax, while CSIOs exhibit a more skeptical stance despite being instructed to advocate for it. This behavior may result from the need for influence campaigns to push less obvious or contradictory content to stimulate discussion or to avoid detection.

## C.3 Interaction Patterns

We analyze the interaction patterns enacted by normal accounts and CSIOs by examining the intra- and intergroup re-shares exchanged by these users. Due to the differing number of users in the two groups, we cannot directly compare the absolute numbers of intra- and intergroup re-shares. Therefore, we normalize the number of interactions by source (i.e., the total number of re-shares that each group performs, see Figure 9). It is important to note that we only consider the re-shares performed and received by normal accounts, as we did not allow CSIO accounts within the same influence campaign to interact with each other by design. Thus, all interaction activities of CSIOs were directed towards normal users.

We observe that normal accounts primarily retweet each other and receive retweets from other normal accounts. This is due to the fact that there are significantly more normal users compared to those operated by CSIO agents. Figure 9 illustrates that normal accounts engage similarly with CSIO accounts from different influence campaigns, except for the GMO campaign, which received relatively fewer re-shares.

## D Influence Analysis

We analyze the follower-following network and measure well-known centrality metrics for both CSIOs and normal users. Figure 10a shows the (Spearman) correlation matrix of the number of influential reports received by the accounts and different centrality metrics, including PageRank, degree centrality, and betweness centrality. For all pairs, we observe moderate correlation but the number of influential reports appears to be less correlated to the other centrality metrics. Figures 10b, 10c and 10d show the distributions of PageRank, degree, and betweenness centrality metrics for normal users and CSIOs. As for the number of influential reports (see Figure 2a in the main paper), we find that these distributions share similar mean but comes from populations with different variances. Specifically, the $p$-values determined according the Brown-Forsythe test are 0.0333, 0.0008, and 0.0074 for page rank, degree centrality and betweenness centrality, respectively.

## D.1 Influence & Activity level

We examine the number of influential reports received by an account as a function of the number of actions that she performed during the experiment. Table 6 reports summary statistics of the regression model fitting the data related to normal users and CSIOs.

## E Discoverability

We define the user's exposure ratio as the ratio between the number of exposure that $u$ had to CSIOs and the total number of exposure. We examine the performance of normal users in detecting CSIOs as a function of their exposure ratio. Table 7 reports summary statistics of the regression model fitting users' precision, recall and F1-score as a function of their exposure ratio.

Received 20 February 2007; revised 12 March 2009; accepted 5 June 2009

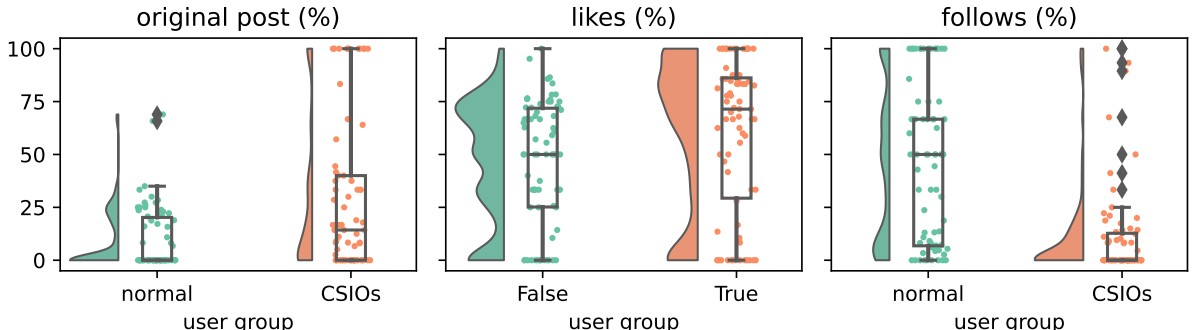

Figure 8: Users activity: The distribution of original posts, likes, follows for normal accounts and CSIOs

Table 4: Most shared hashtags by normal accounts and CSIOs

| Influence Campaign | Covid-19 | Medicare | Wealth Tax | GMO |
|---|---|---|---|---|
| CSIOs | #covidresponse | #healthcareforall | #economicpolicy | #foodsafety |
| | #progress | #healthcarereform | #taxdebate | #healthyliving |
| | #togetherwecan | #thinktwice | #fairness | #informedchoices |
| | #governmentresponse | #doctorssupport | #nowealthtax | #gmoadvocate |
| | #inthistogether | #affordable | #wealthtaxvoices | #gmoheroes |
| | #home | #politics | #thinksmart | #sciencewins |
| | #recovery | #healthyfuture | #supportinnovation | #seralinistudy |
| | #hopeful | #financialfreedom | #fiscalpolicy | #science |
| | #staysafe | #balancedreform | #badpolicy | #gmobenefits |
| | #gratitude | #fairsociety | #debating | #sustainableagriculture |
| Normal | #usagovt | #medicareforall | #economicjustice | #safety |
| | #publichealth | #healthequity | #economicequality | #foodsafety |
| | #pandemic | #universalhealthcare | #people | #research |
| | #futureleaders | #selfcare | #inclusivegrowth | #foodquality |
| | #educationforall | #insurance | #socialgood | #trasparency |
| | #pandemicresponse | #wellness | #taxfairness | #foodsecurity |
| | #safety | #inclusivehealthcare | #taxreform | #nutrition |
| | #innovation | #government | #publicfunding | #futurefood |
| | #virus | #medicalaid | #prosperityforall | #sustainability |

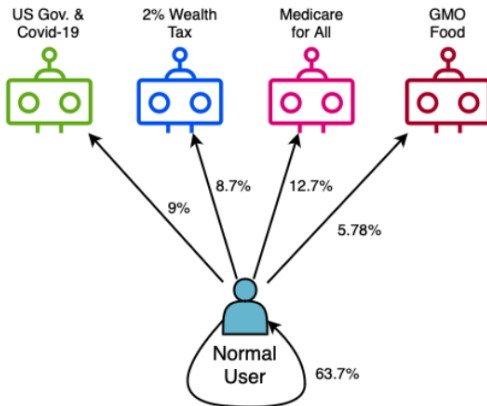

Figure 9: Interaction patterns enacted by normal accounts and CSIOs: proportion of interactions between normal accounts and CSIOs normalized by the source of the re-share

Table 6: Regression analysis' results: number of influential reports received by normal users and CSIOs as a function of their activity level.

| | Normal Users | CSIOs |
|---|---|---|
| $R^2$ | 0.413 | 0.034 |
| adj-$R^2$ | 0.401 | 0.020 |
| $F$-statistic | 34.46 | 2.526 |
| $P(F$-statistic$)$ | 3.71e-7 | 0.116 |
| no. actions (coef.) | 0.128 | 0.024 |
| no. actions (CI) | [0.084, 0.171] | [-0.006,0.056] |
| no. actions ($p$-value) | < 0.0001 | 0.116 |

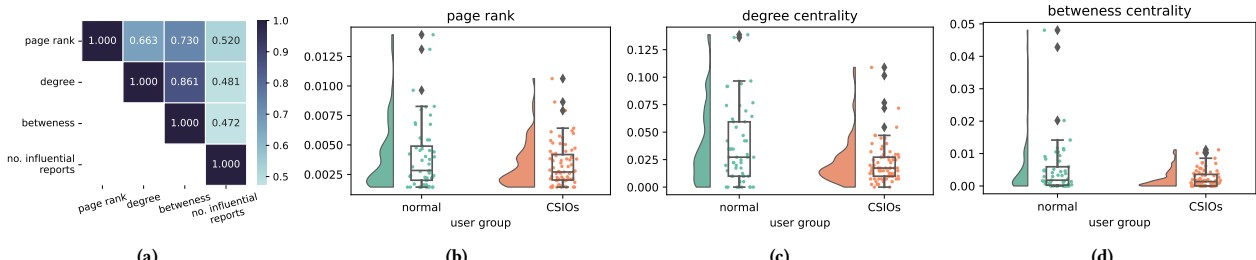

(a)                                    (b)                                    (c)                                    (d)

**Figure 10: Influence analysis: (a) Spearman correlation matrix between the number of influential reports received by a user and his page rank score, his degree centrality and his betweness centrality; (b), (c) and (d) distributions of page rank, degree centrality and betweness centrality, respectively, for normal users and CSIOs.**

**Table 7: Regression analysis' results: precision, recall and F1-score as a function of the exposure ratio.**

|  | Precision | Recall | F1-score |
|---|---|---|---|
| $R^2$ | 0.550 | 0.432 | 0.548 |
| adj-$R^2$ | 0.541 | 0.421 | 0.538 |
| $F$-statistic | 59.92 | 37.29 | 59.31 |
| $P(F\text{-statistic})$ | 4.77e-10 | 1.60e-7 | 5.48e-10 |
| exp. ratio (coef.) | 1.056 | 1.525 | 1.254 |
| exp. ratio (CI) | [0.782, 1.331] | [1.023, 2.027] | [0.927, 1.582] |
| exp. ratio ($p$-value) | < 0.0001 | < 0.0001 | < 0.0001 |

