# OpenReview forum: "Reinforcement-Learning Based Covert Social Influence Operations"
_ACM.org/TheWebConf/2025/Conference — WWW 2025 Poster_

### Official Review · Reviewer_skyo · 2024-12-01

**Novelty:** 5
**Technical Quality:** 4

**Review:**

The authors presented RL_CSIO, a reinforcement-learning-based method to run covert social influence operations.  RL_CSIO studies the trade-offs that CSIO agents face between maximizing influence and minimizing detectability. They ran an IRB-approved 5-day experiment with 225 human subjects using a virtual social media platform and 4 RL_CSIO-driven influence campaigns. With the collected data, they explored 8 research questions related to RL_CSIO-based bots.

Here are the strengths and weaknesses of the paper:

S1: The paper is organized well, and the writing is clear. It is very easy to follow due to the clearly defined concepts.

S2: The authors are worthy of commendation for conducting an IRB-approved 5-day experiment utilizing a virtual social media platform, involving 225 human subjects.

W1: Due to the adoption of a random forest classifier as the bot detector, which may exhibit deviations from the detection strategies employed by bot detectors on social platforms such as X or Facebook, and given the insufficient number of human subjects in the experiment, the universal applicability of the conclusions drawn in this paper may be considered relatively low.

W2: Although I did not carefully check the paper for typos and formatting issues, during my reading, I found 3 issues where the content exceeded the boundaries of the paper's scope and 1 spelling error specifically "platfom" located on line 909. The authors are advised to double-check the paper for errors and make necessary corrections.

**Questions:**

Q1: Did the authors consider how much difference exists between the attributes of the 225 human subjects such as occupation, gender, hobbies, etc. and the attributes of social platform internet users such as Facebook or X?

**Reviewer Confidence:**

3: The reviewer is confident but not certain that the evaluation is correct

**Scope:**

4: The work is relevant to the Web and to the track, and is of broad interest to the community

---

### Official Review · Reviewer_oihp · 2024-12-03

**Novelty:** 3
**Technical Quality:** 3

**Review:**

This paper introduces RL_CSIO, a reinforcement learning-based methodology for running covert social influence operations (CSIOs), exploring the trade-offs between maximizing influence and minimizing detectability. The study is notable for its experimental setup involving 225 human subjects on a virtual social media platform and its insights into bot-human interactions. While the paper provides valuable findings, there are several areas where clarity, depth, and presentation could be improved.
-The abstract is too brief and does not adequately communicate the motivation, methodology, or significance of the work. A more comprehensive summary would make the contributions clearer.
-The introduction does not sufficiently detail the proposed approach or provide the necessary context for readers to grasp the importance and novelty of the work. Including a high-level overview of RL_CSIO and its contributions would enhance the introduction.
-The current version of Figure 1 is not intuitive. Enlarging the figure and adding annotations or a step-by-step explanation would make the framework easier to understand.
-The paper does not include an ablation study to validate the importance of reinforcement learning and graph-based modeling in the proposed framework. Such an analysis would strengthen the argument for using these techniques.

**Questions:**

1. What specific challenges in CSIO modeling does RL_CSIO address that previous approaches could not?
2. How were the research questions framed in the context of the experimental setup, and how do they guide the interpretation of results?
3. What was the rationale for the choice of the 225 human participants, and how representative is the sample?
4. How do the results generalize to real-world social media platforms with more complex interaction dynamics?
5. Could you elaborate on the computational efficiency of RL_CSIO and its scalability for larger datasets or more extended campaigns?

**Reviewer Confidence:**

3: The reviewer is confident but not certain that the evaluation is correct

**Scope:**

2: The connection to the Web is incidental, e.g., use of Web data or API

---

### Official Review · Reviewer_LQdt · 2024-12-03

**Novelty:** 5
**Technical Quality:** 6

**Review:**

The paper performs a 5-day experiment involving human participants and Covert Social Influence Operations (CSIO) agents to investigate how bots interact with humans, how and whether influence their stances, and evade detection. For the experiment execution, the authors propose a reinforcement learning-based methodology, which allows CSIO agents to dynamically adapt bot behaviour in response to feedback (e.g., bots being blocked).

**Pros**
- It is an interesting study that allows the extraction of useful knowledge.
- The experimental methodology is presented clearly, including most of the details necessary for comprehension.
- It is a very well written paper, which allows the reader to clearly understand the findings and conclusions.

**Cons**
- I believe the paper would benefit from a dedicated discussion of its limitations and potential applications of the insights. Specifically, how can social media platform administrators leverage these findings to improve user experience and mitigate bot-based manipulation?

&nbsp;

**Other comments**
- It would be nice if the authors provided information about the process followed to generate the GPT3.5-based posts mentioned in Section 3.2.1.
- What was the initial number of people recruited through Amazon Mechanical Turk before they were disqualified for not passing the necessary checks?
- Please ensure all abbreviations are introduced with their full form upon first mention, e.g., RL and IRB in the Abstract.

**Questions:**

1. How can social media platform administrators use the findings of this study to improve user experience?
2. What are the limitations of this work from the author's perspective?

**Reviewer Confidence:**

3: The reviewer is confident but not certain that the evaluation is correct

**Scope:**

4: The work is relevant to the Web and to the track, and is of broad interest to the community

---

### Official Review · Reviewer_Uc6R · 2024-12-04

**Novelty:** 5
**Technical Quality:** 5

**Review:**

This paper presents a reinforcement learning (RL)-based framework, RL_CSIO, for managing covert social influence operations (CSIOs) on social media. The study focuses on balancing two competing objectives: maximizing influence on human participants and minimizing the detectability of bot accounts. The work stands out for its innovative application of RL to this challenging and impactful problem, providing fresh insights into the dynamics of covert influence campaigns. The paper also emphasizes its ethical rigor by running IRB-approved experiments with human participants on a controlled platform, which enhances the credibility of its findings.

Strengths:

1. The paper leverages reinforcement learning to dynamically balance influence and detectability, a fresh perspective in the field. The proposed RL_CSIO framework demonstrates adaptability and potential for practical application.

2. The study employs an IRB-approved experiment using a virtual platform and real participants, ensuring ethical rigor and realistic data collection. The comprehensive evaluation metrics, including influence, stance change, and detectability, provide a well-rounded assessment of the methodology.

3. The study highlights significant implications for platform moderation and societal discourse, offering insights that could inform regulatory strategies and countermeasures.

Weakness:

1. The dataset is relatively small (225 participants) and unevenly distributed, which may limit the generalizability of the results. A discussion of this limitation and potential remedies (e.g., simulations or larger-scale experiments) would be valuable.

2. The analysis of the relationship between influence and activity levels is somewhat inconsistent (e.g., weaker correlation for bots compared to humans), and further explanation is warranted.

**Questions:**

See the weakness

**Reviewer Confidence:**

3: The reviewer is confident but not certain that the evaluation is correct

**Scope:**

4: The work is relevant to the Web and to the track, and is of broad interest to the community